# Silica-Coated Magnetic Iron Oxide Nanoparticles Grafted onto Graphene Oxide for Protein Isolation

**DOI:** 10.3390/nano10010117

**Published:** 2020-01-08

**Authors:** Xuan-Hung Pham, Eunil Hahm, Hyung-Mo Kim, Byung Sung Son, Ahla Jo, Jaehyun An, Tuong An Tran Thi, Dinh Quan Nguyen, Bong-Hyun Jun

**Affiliations:** 1Department of Bioscience and Biotechnology, Konkuk University, Seoul 05029, Korea; phamricky@gmail.com (X.-H.P.); greenice@konkuk.ac.kr (E.H.); hmkim0109@konkuk.ac.kr (H.-M.K.); imsonbs@konkuk.ac.kr (B.S.S.); iamara0421@konkuk.ac.kr (A.J.); ghj4067@konkuk.ac.kr (J.A.); 2Laboratory of Biofuel and Biomass Research, VNU-HCMU University of Technology, 268 Ly Thuong Kiet street, district 10, Ho Chi Minh 700000, Vietnam; trantuongan81@gmail.com (T.A.T.T.); ndquan@hcmut.edu.vn (D.Q.N.)

**Keywords:** graphene oxide, superparamagnetic iron oxide nanoparticles, reverse microemulsion, silica-coated superparamagnetic nanoparticle, superparamagnetic nanoparticle embedded graphene oxide, protein isolation

## Abstract

In this study, silica-coated magnetic iron oxide nanoparticles (MNPs@SiO_2_) were covalently conjugated onto graphene oxide (GO/MNP@SiO_2_) for protein isolation. First, MNPs were precisely coated with a silica layer on the surface by using the reverse microemulsion method, followed by incubation with 3-glycidyloxypropyltrimethoxysilane (GPTS) to produce the GPTS-functionalized MNPs@SiO_2_ (GPTS-coated MNPs@SiO_2_) that display epoxy groups on the surface. The silica shell on the MNPs was optimized at 300 µL of Igepal^®^CO-520, 5 mg of MNP, 100 µL of TEOS, 100 µL of NH_4_OH and 3% of 3-glycidyloxypropyltrimethoxysilane (GPTS). Simultaneously, polyethyleneimine (PEI) was covalently conjugated to GO to enhance the stability of GO in aqueous solutions and create the reaction sites with epoxy groups on the surface of GPTS-coated MNP@SiO_2_. The ratio of PEI grafted GO and GPTS-coated MNP@SiO_2_ (GO/MNP ratio) was investigated to produce GO/MNPs@SiO_2_ with highly saturated magnetization without aggregation. As a result, the GO/MNP ratio of 5 was the best condition to produce the GO/MNP@SiO_2_ with 9.53 emu/g of saturation superparamagnetization at a magnetic field of 2.0 (T). Finally, the GO/MNPs@SiO_2_ were used to separate bovine serum albumin (BSA) to investigate its protein isolation ability. The quantity of BSA adsorbed onto 1 mg of GO/MNP@SiO_2_ increased sharply over time to reach 628 ± 9.3 µg/mg after 15 min, which was 3.5-fold-higher than that of GPTS-coated MNP@SiO_2_. This result suggests that the GO/MNP@SiO_2_ nanostructure can be used for protein isolation.

## 1. Introduction

Graphene oxide (GO) has attracted significant attention in recent decades due to its attractive properties, such as its large surface area, high electron transport capability, elasticity, thermal conductivity, mechanical strength, and tunable optical properties [1]. As a result, graphene oxide is utilized in many applications, such as catalysis [2], sensing [3], energy [4], drug delivery [5], and biomedical applications—deep brain stimulators [6], blood glucose sensors [7], tissue engineering [8,9], gene therapy [10], cell imaging [11], cancer therapies [12], and to differentiate and image stem cells [13]. However, the separation of GO from solution is a time-consuming and complicated process [14].

Magnetic iron oxide nanoparticles (MNPs) with diameters of 20 nm or less are superparamagnetic [15] and have been attracted due to their superparamagnetism, high field irreversibility, high saturation field, extra anisotropy contributions or shifted loops after field cooling non-toxicity and biocompatibility [16,17]. As a result, the MNP was applied in various biomedical applications such as drug delivery [18,19], chemotherapy [17] and bioimaging [15]. There are several routes for chemical synthesis of superparamagnetic MNP: co-precipitation, microemulsion, sonochemical, hydrothermal and so on [17]. Among these methods, the synthesis of MNP by coprecipitation is facile and can be easily scaled up. However, the size of MNP is widely distributed and agglomerated because of Oswald ripening [17]. The MNP synthesized by microemulsion (water-in-oil) is formed inside of water droplets and can be well controlled in size by controlling the size of these water droplets in microemulsion system. However, these NPs may not be useful for drug delivery purposes [17]. In addition, the synthesis of MNP by sonochemical was limited by large-scale synthesis [17]. The hydrothermal method is reported to produce MNPs of uniform sizes and can easily be scaled up [17]. However, the magnetic force of each single MNP is too weak under the externally applied magnetic field and limiting their practical application [15,17,20,21].

Recently, the combination of GO with nanoparticles, forming graphene–nanoparticle hybrid structures, has been shown to offer many additional unique physicochemical properties that are markedly advantageous for applications, such as supercapacitors, electrode materials, drug carriers, and bioimaging [22]. Among these graphene-nanoparticle hybrid materials, MNP-decorated GO nanocomposites (GO-MNP) have been synthesized by several groups for removing hazard materials [23,24,25], sensing/biosensing [26,27,28,29,30], enrichment and adsorption of peptide or protein [31,32], sample manipulation for bio-separation [30,33], bioimaging [34] and drug delivery [35,36] applications. GO-MNP nanocomposites are formed through in situ reduction of iron salt precursors [37,38] or assembly of the MNPs on the GO surface [39,40]. The former approach, however, is challenging, as there is minimal control over the size, size distribution, and location of MNPs on the GO sheets. The latter approach enables well-controlled size, size distribution, morphology, and controlled decoration ratio on the edge of the GO sheets [40]. Ye et al. prepared GO/MNP nanocomposite by simply drop-wise adding MNP into GO solution through electrostatic interaction [31]. The covalent conjugation of MNP and GO were reported for preparing GO/MNP using “click” chemistry by Zhang et al. [23]. The surface modification of MNP was; however, complicated. In addition, Zhang et al. covalently bound MNPs to a reduced GO sheet by modifying the MNP with hydrophilic 2,3-dimercaptosuccinnic acid (DMSA), followed by assembly onto GO-grafted PEI sheets [40]. However, the mechanism of DMSA adsorption onto the MNP remains unclear. In addition, the GO-MNP composites agglomerated, resulting in gradual precipitation from their aqueous suspension [35].

Recently, silica-coated magnetic iron oxide MNPs (MNPs@SiO_2_) have been successfully developed and used in biomedical applications, such as macromolecular separation [41]. The silica layer not only improves the stability in aqueous solution, but also facilitates surface modification with different functional groups [42,43]. MNPs@SiO_2_ can be prepared by several methods, such as Stöber [44] or reverse microemulsion [45]. However, the magnetic properties of individual MNPs@SiO_2_ are weak due to an excess of amorphous silica that lowers the individual magnetic force, thereby limiting their practical application [20,21]. To overcome the weak magnetic force of MNP@SiO_2_ and the difficult separation of GO, we develop a new nanomaterial by conjugating MNP@SiO_2_ to GO platform by covalent bonding in this study. A reverse microemulsion method was used to control the silica coating on the MNPs to produce MNPs@SiO_2_ with highly saturated magnetization. MNPs@SiO_2_ were covalently bound to GO to form the MNP functionalized GO. The presence of GO as a platform for conjugating to MNPs@SiO_2_ facilitates the isolation and separation of biomaterial from solution_._ In this process, PEI was first covalently conjugated to GO to enhance the stability of GO in aqueous solutions. MNPs were precisely coated with a silica layer on the surface, followed by incubation with 3-glycidyloxypropyltrimethoxysilane (GPTS) to produce the GPTS-functionalized MNPs@SiO_2_ (GPTS-coated MNPs@SiO_2_) that display epoxy groups on the surface. GPTS-coated MNPs@SiO_2_ were conjugated onto the surface of PEI grafted GO to produce GO/MNPs@SiO_2_ with highly saturated magnetization. Finally, the GO/MNPs@SiO_2_ were used to separate bovine serum albumin (BSA) to investigate its protein isolation ability.

## 2. Materials and Methods 

### 2.1. Chemicals and Materials

Graphene oxide (GO) was purchased from Graphene Laboratory Inc. (Ronkonkoma, NY, USA). Cyclohexane, Igepal^®^CO-520 (Igepal, M_n_ = 441), branched polyethyleneimine (PEI, 25k), N-Hydroxysuccinimide (NHS), N-(3-Dimethylaminopropyl)-N′-ethyl carbodiimide hydrochloride (EDC), tetraethyl orthosilicate (TEOS), 3-Glycidyloxypropyltrimethoxysilane (GPTS), ethyl alcohol (EtOH), bovine serum albumin (BSA) and phosphate buffer saline (PBS) were purchased from Sigma-Aldrich (St. Louis, MO, USA) and used without further purification. Fe_3_O_4_ nanoparticles (MNP) with superparamagnetic property were purchased from Ocean Nanotech (San Diego, CA, USA). Aqueous ammonium hydroxide (NH_4_OH, 27%) was purchased from Daejung (Siheung, Korea). DI water was purified using a Direct-Q Millipore water purification system (Sam Woo S&T Co., Ltd., Seongnam, Gyeonggi-do, Korea.)

### 2.2. Characterization

Transmission electron microscope (TEM) images of the sample were taken using a Libra 120 field-emission transmission electron microscope (Carl Zeiss, German) with a maximum accelerated voltage of 120 kV. Optical properties of our materials were observed by an Optizen POP UV/Vis Spectrometer (Mecasys, Seoul, Korea). Magnetic measurement was carried out using a PPMS-9 vibrating sample magnetometer (Quantum Design, San Diego, CA, USA) at room temperature. Attenuated total reflection-Fourier transform infrared spectra of our materials was recorded on a Nicolet^TM^ iS^TM^ 50 Fourier transform infrared spectroscopy (FTIR) Spectrometer (Thermo Fisher, Waltham, MA, USA).

### 2.3. Optimized Preparation of Silica-Coated Magnetic Fe_3_O_4_ Nanoparticles

#### 2.3.1. Effect of Igepal^®^CO-520 Volume on Silica-Coated Magnetic Fe_3_O_4_ Nanoparticles

MNPs were coated with silica by the reverse microemulsion method. Various volumes of Igepal^®^CO-520 (0.1 to 5 mL) were added into 11 mL cyclohexane and sonicated for 10 min. Subsequently, 5 mg of MNPs in chloroform was added to the solution and mixed for several minutes to disperse the MNPs into the Igepal^®^CO-520 solution. Next, NH_4_OH (100 µL) and TEOS (100 µL) were added drop-wise, and the mixture was incubated for 16 h at 25 ℃. Next, the mixture was precipitated by adding 20 mL EtOH and centrifuged at 8500 rpm for 30 min. The precipitate was washed five times with EtOH. Finally, the MNPs@SiO_2_ were re-dispersed into EtOH to obtain a 1 mg/mL MNP@SiO_2_ solution.

#### 2.3.2. Effect of Quantity of MNPs on Silica-Coated Magnetic Fe_3_O_4_ Nanoparticles

A volume of 0.3 mL of Igepal^®^CO-520 was added into 11 mL cyclohexane and sonicated for 10 min. Subsequently, various quantities of MNPs in chloroform (1, 3, 5, 7 and 10 mg) were added to the solution and mixed for several minutes to disperse the MNPs into the Igepal^®^CO-520 solution. Next, NH_4_OH (100 µL) and TEOS (100 µL) were added drop-wise, and the mixture was incubated for 16 h at 25 ℃. Next, the mixture was precipitated by adding 20 mL EtOH and centrifuged at 8500 rpm for 30 min. The precipitate was washed five times with EtOH. Finally, the MNPs@SiO_2_ were re-dispersed into EtOH to obtain a 1 mg/mL MNP@SiO_2_ solution.

#### 2.3.3. Effect of TEOS Volume on Silica-Coated Magnetic Fe_3_O_4_ Nanoparticles

A volume of 0.3 mL of Igepal^®^CO-520 was added into 11 mL cyclohexane and sonicated for 10 min. Subsequently, 5 mg of MNPs in chloroform was added to the solution and mixed for several min to disperse the MNPs into the Igepal^®^CO-520 solution. Next, NH_4_OH (100 µL) and various volumes of TEOS (10, 50, 100, 150 and 200 µL) were added drop-wise, and the mixture was incubated for 16 h at 25 °C. Next, the mixture was precipitated by adding 20 mL EtOH and centrifuged at 8500 rpm for 30 min. The precipitate was washed five times with EtOH. Finally, the MNPs@SiO_2_ were re-dispersed into EtOH to obtain a 1 mg/mL MNP@SiO_2_ solution.

#### 2.3.4. Effect of GPTS Concentration on Silica-Coated Magnetic Fe_3_O_4_ Nanoparticles

MNPs@SiO_2_ were re-dispersed into dimethylfuran (DMF) to obtain a 1 mg/mL MNP@SiO_2_ solution. Epoxy groups were introduced onto the surface of MNPs@SiO_2_ by incubating with various concentrations of GPTS (1, 2, 3, 5%) in DMF for 1 h. GPTS-coated MNPs@SiO_2_ were collected by centrifuging at 15,000 rpm for 30 min, washed several times, and dispersed into DMF to obtain a 5 mg/mL GPTS-coated MNP@SiO_2_ solution. 

### 2.4. Preparation of Graphene Oxide Grafted with Polyethyleneimine (PEI grafted GO)

GO (2.5 mg in 0.5 mL H_2_O) was added to a PEI solution (400 µg PEI in 9.5 mL H_2_O) to obtain a final concentration of 40 µg/mL of PEI. The mixture was sonicated for 30 min and incubated at 60 °C for 12 h. Then, the mixture was centrifuged at 17,000 rpm for 1 h, and the precipitate was washed five times with PBS solution to remove the unbound PEI. PEI-adsorbed to GO was re-dispersed in a 0.5 mL PBS.

EDC (20 mg) and NHS (14 mg) in 0.5 mL PBS were added into the PEI-adsorbed GO solution to activate the carboxyl groups for 1 h. PEI (20 mg) was added to the mixture. The mixture was incubated for 2 h at room temperature to conjugate PEI onto the GO surface. An ethanolamine solution (100 µL, 1 M) was added to the mixture and incubated for 1 h to block the unreacted carboxyl groups. PEI grafted GO was collected by centrifugation at 17,000 rpm for 30 min and re-dispersed in 5 mL PBS.

### 2.5. Preparation of Silica-Coated Magnetic Nanoparticles Grafted onto Graphene Oxide (GO/MNP@SiO2)

GPTS-coated MNP@SiO_2_ solution (250 µg) was added to the PEI-grafted GO solution (250 to 2500 µg) and vigorously mixed for 2 h at 25 °C. An ethanolamine solution (100 µL, 1.0 M) was added to the mixture and incubated for 1 h to block the unreacted epoxy groups. GO/MNP@SiO_2_ was collected by centrifugation 3 times at 8500 rpm for 15 min, followed by applying a magnetic force for 5 min and washing 5 times with PBS. 

### 2.6. BSA Isolation by GO/MNP@SiO_2_


BSA was adsorbed into GO/MNP@SiO_2_ by incubation. Briefly, 1 mg of GO/MNP@SiO_2_ in PBS was incubated with 1% BSA in PBS buffer (100 µL) for 2 h. BSA adsorbed GO/MNP@SiO_2_ was collected by applying a magnetic separator. The BSA adsorbed GO/MNP@SiO_2_ was washed several times with PBS, and the supernatant was collected and diluted to 50 mL. The concentration of BSA in supernatant was determined by UV–Vis spectroscopy at 280 nm. Similarly, MNP@SiO_2_-GPTS was used as a control sample for BSA isolation.

## 3. Results and Discussion

Superparamagnetic MNPs (~20 nm) were coated with a silica layer using the reverse microemulsion method. The surface of MNP@SiO_2_ was modified with epoxy groups by incubating them with GPTS (MNPs@ SiO_2_-GPTS). Simultaneously, PEI was first adsorbed on the GO surface for 24 h at 60 °C, and stabilized the GO surface so that GO can be well dispersed in an aqueous solution (PEI adsorbed GO). Subsequently, the COOH groups of the GO surface were coupled with the NH_2_ groups of PEI using EDC/NHS (PEI grafted GO). Finally, GPTS-coated MNPs@SiO_2_ were conjugated to the surface of PEI grafted GO by coupling the PEI NH_2_ groups with the epoxy groups on the surface of GPTS-coated MNP@SiO_2_, producing GO/MNP@SiO_2_ nanocomposites (Figure 1).

### 3.1. Optimized Preparation of Silica-Coated Magnetic Nanoparticles (MNPs@SiO_2_)

The coating of the silica layer on the MNP surface via the reverse microemulsion method was optimized by varying experimental parameters such as Igepal CO-520 and TEOS volumes, as well as the number density of MNPs.

#### 3.1.1. Effect of Igepal^®^CO-520 Volume on Silica-Coated Magnetic Fe_3_O_4_ Nanoparticles

In this study, the average diameter of superparamagnetic MNPs is 20.6 ± 1.0 nm (n = 20) as determined by ImageJ software analysis on TEM images (Figure 2a(i)). Figure 2a(ii–viii) shows TEM images of MNPs@SiO_2_ synthesized with varying Igepal^®^CO-520 volumes (100 to 5000 µL). At a low surfactant volume (100 µL), aggregated core-free silica nanoparticles and superparamagnetic MNP@SiO_2_ were formed in Figure 2a(ii). According to the previous report, Igepal^®^CO-520 self-assembles into micelles in cyclohexane due to its hydrophilic groups (Appendix A). The oleic acid that adsorbed on the surface of superparamagnetic MNP during synthesis was exchanged to Igepal^®^CO-520 in the cyclohexane [46]. Therefore, at low volume Igepal^®^CO-520 cannot self-assemble into micelles for MNP@SiO_2_ formation. At volume that concentration is higher than the surfactant CMC (200 to 500 µL), superparamagnetic MNPs@SiO_2_ were well obtained with diameters of 30.2 ± 1.7, 37.7 ± 1.4, 37.4 ± 1.5, and 32.1 ± 1.8 nm with 200, 300, 400, and 500 µL of surfactant, respectively (Figure 2b). However, Igepal^®^CO-520 micelles could fuse at extremely high volumes (>2500 µL) [47], resulting in aggregated MNPs@SiO_2_ and core-free silica nanoparticles (bare SiO_2_ NPs). As a result, 300 µL of Igepal^®^CO-520 was applied for further study.

#### 3.1.2. Effect of Quantity of MNPs on Silica-Coated Magnetic Fe_3_O_4_ Nanoparticles

To obtain superparamagnetic MNPs@SiO_2_ without the presence of bare SiO_2_ NPs, the ratio of MNP and micelle number density is a critical parameter. Varying the MNP quantity (1 to 10 mg) revealed that bare SiO_2_ NPs were produced with 1 mg of MNPs (Figure 3). The diameter of MNPs@SiO_2_ when 1 mg of MNP was used equaled 46.3 ± 1.1 nm, which is larger than the diameter of 37.7 ± 1.4 nm when 5 mg of MNP was used. This result indicates that the silica layer of MNPs is thicker and TEOS was excess in solution at 1 mg of MNP than at 5 mg of MNP. Along with the disappearance of the bare SiO_2_ NPs, the diameter of MNPs@SiO_2_ decreased with increasing quantity of MNP (Figure 3c). However, at high quantity of MNP at 7 mg or 10 mg, the TEOS in solution was not enough to produce the silica layer on the surface of all MNPs. MNP@SiO_2_ was became irregular shape (Figure 3a). Therefore, 5 mg of MNPs was chosen for further study.

#### 3.1.3. Effect of TEOS Volume on Silica-Coated Magnetic Fe_3_O_4_ Nanoparticles

The volume of TEOS precursor is an important factor that affected on the thickness of silica shell of superparamagnetic MNP and the formation of bare SiO_2_ NP formation in the solution [46,48]. When TEOS is then added to the mixture, it converts to hydrolyzed TEOS adsorbed onto the superparamagnetic MNP surface, which results in transfer of the superparamagnetic MNPs into the water phase [46]. Finally, the hydrolyzed TEOS on the superparamagnetic MNP surface undergoes a condensation reaction to form the silica shell [46]. As the TEOS content increased from 10 to 100 μL, the diameter of MNPs@SiO_2_ and thickness of silica shell on the surface of MNP increased. It reached a plateau at 100 µL of TEOS in Figure 4. The silica layer was negligible at 10 µL, while the silica layer was not uniform at 50 µL of TEOS. The MNP@SiO_2_ with uniform thickness of silica layer on the surface was obtained at 100 µL of TEOS. The MNP@SiO_2_ diameter increased to 24.2 ± 2.0, 31.1 ± 2.3, and 37.7 ± 1.4 nm with 10, 50, and 100 µL of TEOS, respectively (Figure 4b). Increasing the volume of TEOS further caused the formation of bare SiO_2_ NPs, while the diameter of MNP@SiO_2_ was insignificantly different (Figure 4b,c). So, 100 µL of TEOS was chosen for further study.

#### 3.1.4. Effect of GPTS Concentration on Silica-Coated Magnetic Fe_3_O_4_ Nanoparticles

To conjugate the MNPs@SiO_2_ onto the GO-grafted PEI, superparamagnetic MNPs@SiO_2_ must be modified with GPTS to create GPTS-coated MNPs@SiO_2_ that displayed epoxy groups on the surface, enabling conjugation with the –NH_2_ groups of PEIs. The superparamagnetic MNPs@SiO_2_ were incubated in a toluene solution containing 1 to 3% GPTS for 1 h. The diameter remained unchanged and no bare SiO_2_ was obtained after GPTS treatment (Appendix A). So, 3% GPTS was chosen for further study.

### 3.2. Preparation of Graphene Oxide Grafted Polyethyleneimine (GO-grafted PEI)

GO is utilized as a platform structure for conjugating to GPTS-coated MNP@SiO_2_ in our study. However, GO was easily aggregated in aqueous solution. Therefore, GO was dispersed in solutions of PEI in our study to improve it stability aqueous solution [49,50,51]. GO was incubated with PEI to enables dispersion of GO in aqueous solutions due to its intrinsic hydrophilicity [52]. Therefore, two sizes of branched PEI (molecular weights of 600 (PEI 600), and 25,000 (PEI 25k) were investigated for GO grafting. In Appendix A, GO was dispersed well in PEI 25k but aggregated in PEI 600, as seen in the optical images. This suggests that the low molecular weight of PEI 600 cannot stabilize and dispersed GO in aqueous solutions. TEM images and the absorbance spectra of PEI 25k grafted GO are shown in Appendix A. PEI grafted GO displays a smoother and flatter surface than bare GO. The absorbance spectra of PEI grafted GO shows a significant increase in absorbance intensity in the UV region (Appendix A), indicating the presence of PEI on the surface of GO. 

To increase the linking chain position on the surface, GO was first conjugated with PEI to modify the GO surface with –NH_2_ groups due to its excellent cationic polymer [53,54,55]. To confirm that PEI 25k is covalently bound to GO, we used attenuated total reflection-Fourier transform infrared spectroscopy (ATR-FTIR) to characterize the functional groups of PEI grafted GO. FTIR spectra of GO, and PEI 25k grafted GO, are shown in Appendix A. The presence of various oxygen functionalities is present in the GO sample. A broadband and high absorbance intensity from 3000–3700 cm^−1^ was assigned to the stretching of the hydroxyl group (–OH) [56]. Narrow intense peaks at 1715 cm^−1^ and 1621 cm^−1^ represent characteristic stretching of –COO^–^ groups and C=C vibration of aromatic chains on the GO surface. GO also showed peaks at 1352, 1216, 1036, and 974 cm^−1^ representing the carboxyl/carbonyl, epoxy, and alkoxy groups, respectively [14,56,57,58].

In contrast, PEI grafted GO exhibited additional peaks at 3100, 2928, and 2821 cm^−1^, which are attributed to the stretching of amide B and stretching and bending of –CH_2_ from PEI. In addition, the disappearance of the 1715 cm^−1^ peak and occurrence of new peaks at 1640 cm^−1^ (Amide I) and 1544 cm^−1^ (Amide II) indicate that the carboxyl groups reacted with the primary amine groups to generate the amide bonds. This result indicates that the conjugation of PEI to the GO surface was successful.

### 3.3. Conjugation of Silica-Coated Magnetic Nanoparticles on Graphene Oxide Grafted Polyethyleneimine 

PEI grafted GO was reacted with GPTS-coated MNPs@SiO_2_ by incubating 250 µg of GPTS-coated MNPs@SiO_2_ with PEI grafted GO (250 to 2500 µg) at weight-to-weight ratios (GO/MNP) of 2:1, 3:1, 4:1, 5:1, 6:1, 8:1, and 10:1 (Figure 5a and Appendix A). At a low GO/MNP ratio, GPTS-coated MNPs@SiO_2_ were aggregated on the GO surface and precipitated. At the highest GO/MNP ratio (10:1), the density of MNPs on the GO surface was too low to impart magnetic properties. Overall, the surface density of GPTS-coated MNPs@SiO_2_ decreased with increasing GO content. Therefore, a GO/MNP ratio of 5:1 was chosen for subsequent studies.

The conjugation of GPTS-coated MNPs@SiO_2_ onto PEI-grafted GO was confirmed by determining the functional groups with ATR-FTIR (Figure 5b). The GPTS-coated MNPs@SiO_2_ showed a typical spectrum with peaks at 949 cm^−1^ and 1053 cm^−1^ that are attributed to the oxirane group from the GPTS and Si-O from the silica [59]. The intensity ratio of peaks at 949 to 1053 cm^−1^ is 1.67. When GPTS-coated MNPs@SiO_2_ were conjugated to PEI grafted GO, the peak ratio between 949 to 1053 cm^−1^ decreased to 1.05. This result indicates that the oxirane peak of GPTS-coated MNPs@SiO_2_ reacted with the primary amine groups of PEI. The optical properties of GO/MNP@SiO_2_ nanomaterial was showed in Figure 5c. The UV–Vis spectra of GO/MNP@SiO_2_ showed the combination properties of both GPTS-coated MNP@SiO_2_ with the strong absorbance at ~210, 250 and 350 nm and PEI grafted GO at 280 nm. Therefore, GPTS-coated MNPs@SiO_2_ were successfully conjugated onto the PEI-grafted GO to generate the GO/MNPs@SiO_2_.

### 3.4. The Magnetization and Protein Isolation of Silica-Coated Magnetic Nanoparticles on Graphene Oxide Grafted Polyethyleneimine

The magnetic property of the GO/MNP@SiO_2_ was investigated by using a PPMS-9 vibrating sample magnetometer. The saturation magnetization of the GO/MNPs@SiO_2_ was determined to be 9.53 emu/g at magnetic field of 2.0 (T), which is higher than the magnetization of MNPs in previous reports [40,60] (Figure 6a). The magnetization as the function of applied field showed a reversible S-shape and no remnant magnetization is obtained, indicating a superparamagnetic behavior of the GO/MNP@SiO_2_. The magnetic property of the GO/MNPs@SiO_2_ was dramatically improved compared to that of MNP in the inset of Figure 6a. After applying of magnet for 5 min, the GO/MNPs@SiO_2_ was easily collected, while the MNP@SiO_2_ still dispersed in aqueous solution. It indicated that the presence of GO as the platform for conjugating the MNP@SiO_2_ facility the collection of MNP. 

For application, GO has been shown to be a suitable platform for protein and nucleic acid adsorption [14,61]. However, the separation of GO from solution is a time-consuming and complicated process [14]. As mentioned above, the MNPs@SiO_2_ was easily collected when conjugated on the surface of GO in our study, Therefore, we used the GO/MNPs@SiO_2_ nanocomposites with superparamagnetic property as an isolation material to investigate the isolation of protein in solution. BSA is used as a model protein to investigate the isolation ability of the GO/MNPs@SiO_2_ nanocomposites. The GPTS-coated MNP@SiO_2_ was utilized as a control material in our study. As shown in Appendix A, BSA solution exhibited an absorbance peak at 280 nm because of its protein property. This peak was linearly proportional to the BSA concentration in the range of 0 to 100 µM (Figure 6b). The quantity of BSA that adsorbed onto the nanomaterials in our study was observed according to the incubation time. According to the previous report, GO is an amphiphile with both hydrophilic edges and hydrophobic basal plane [62]. The π–π stacking interaction between tyrosine or tryptophan residues of BSA and hydrophobic basal plane have an important role for BSA absorbed on GO/MNPs@SiO_2_ [62]. Indeed, BSA adsorbed on 1 mg of GO/MNP@SiO_2_ and GPTS-coated MNP@SiO_2_ initially increased sharply and achieved a maximum value in Figure 6b. The capability of adsorbed BSA per mg of GO/MNP@SiO_2_ was 628 ± 9.3 µg/mg after 15 min incubation, adsorbing 65%–69% of the BSA from solution (Figure 6b). The BSA adsorption capability of GPTS-coated MNP@SiO_2_ was 178 ± 154 µg/mg, which was 3.5-fold less than that of the GO/MNP@SiO_2_. This result indicates that BSA can be easily adsorbed onto the GO/MNP@SiO_2_ nanocomposite and collected due to the presence of GO as a platform and MNP as a magnetic separator material. In our previous report, we used graphene oxide conjugated magnetic beads to extract total RNA and miRNA from non-small cell lung cancer cells [14]. Therefore, we believed that it is also possible to isolate RNA and miRNA from a matrix containing sugars, DNA and other biomolecules by using GO/MNP@SiO_2_ nanomaterial. In conclusion, this result suggests that the GO/MNP@SiO_2_ nanostructure can be used for protein and nucleic isolation.

## 4. Conclusions

The silica-coated magnetic iron oxide nanoparticles conjugated to graphene oxide for protein separation were successfully prepared. The silica shell on the superparamagnetic MNPs was optimized by adjusting experimental conditions (300 µL of Igepal^®^CO-520, 5 mg of MNP, 100 µL of TEOS, and 100 µL of NH_4_OH in cyclohexane) to produce MNPs@SiO_2_ with highly saturated magnetization. MNPs@SiO_2_ were incubated with GPTS to produce GPTS-coated MNPs@SiO_2_ with surface functionalized epoxy groups. GO was incubated and covalently conjugated with PEI to enhance the stability of GO in aqueous solution and introduce reactive functional groups. GPTS-coated MNPs@SiO_2_ were conjugated onto the surface of GO-grafted PEI at a GO/MNP ratio of 5. The saturation superparamagnetization value of the GO/MNP@SiO_2_ nanocomposite was 9.53 emu/g at a magnetic field of 2.0 (T). The GO/MNP@SiO_2_ nanocomposite was used to separate BSA from solution. The quantity of BSA adsorbed onto 1 mg of GO/MNP@SiO_2_ increased sharply and achieved a maximum value of 65%–69% after 15 min. This result suggests that the preparation method for GO/MNP@SiO_2_ can be used to fabricate nanocomposites for protein isolation.

## Figures and Tables

**Figure 1 nanomaterials-10-00117-f001:**
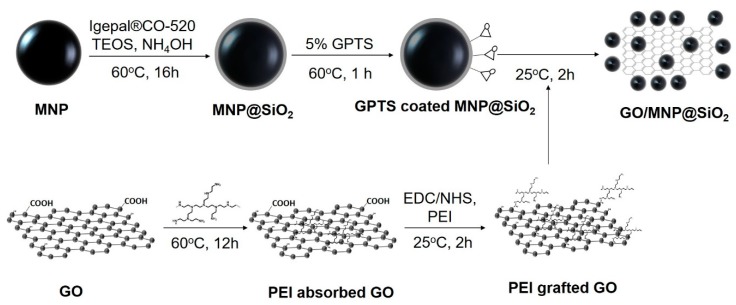
Schematic of the preparation of silica-coated superparamagnetic iron oxide nanoparticles grafted graphene oxide nanostructures (GO/MNP@SiO_2_).

**Figure 2 nanomaterials-10-00117-f002:**
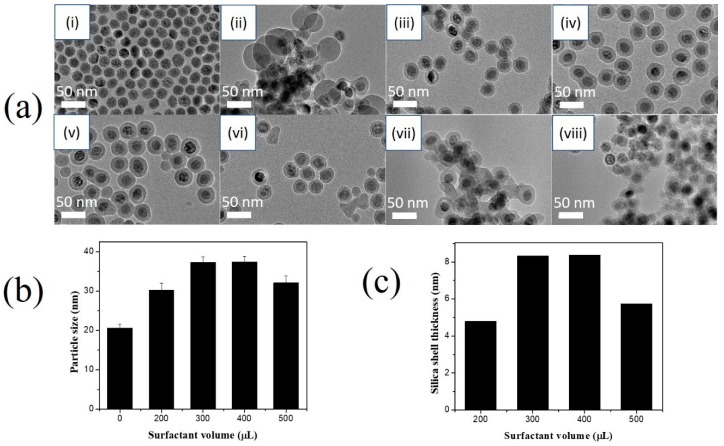
Effects of reverse microemulsion conditions on the silica coating of the magnetic nanoparticles. (**a**) Transmission electron microscope (TEM) images, (**b**) particle size and (**c**) silica shell thickness of MNP@SiO_2_ synthesized at different Igepal^®^CO-520 surfactant volumes: (i) 0 µL, (ii) 100 µL, (iii) 200 µL, (iv) 300 µL, (v) 400 µL, (vi) 500 µL, (vii) 2500 µL, and (viii) 5000 µL in the presence of 5 mg of MNPs, 100 µL of TEOS, and 100 µL of NH_4_OH.

**Figure 3 nanomaterials-10-00117-f003:**
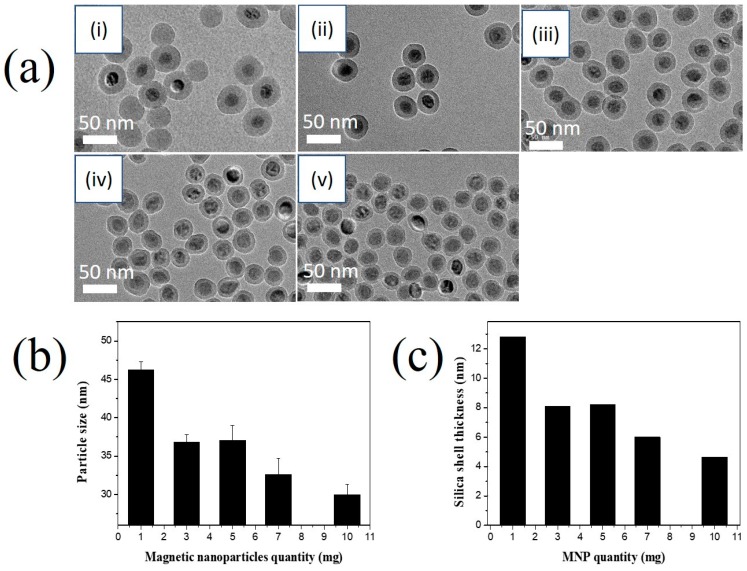
Effects of reverse microemulsion conditions on the silica coating of the magnetic nanoparticles. (**a**) TEM images (**b**) particle size and (**c**) silica shell thickness of MNP@SiO_2_ synthesized at different quantity of MNP: (i) 1 mg, (ii) 3 mg, (iii) 5 mg, (iv) 7 mg, and (v) 10 mg in the presence of 0.3 mL of Igepal^®^CO-520, 100 µL of TEOS, and 100 µL of NH_4_OH.

**Figure 4 nanomaterials-10-00117-f004:**
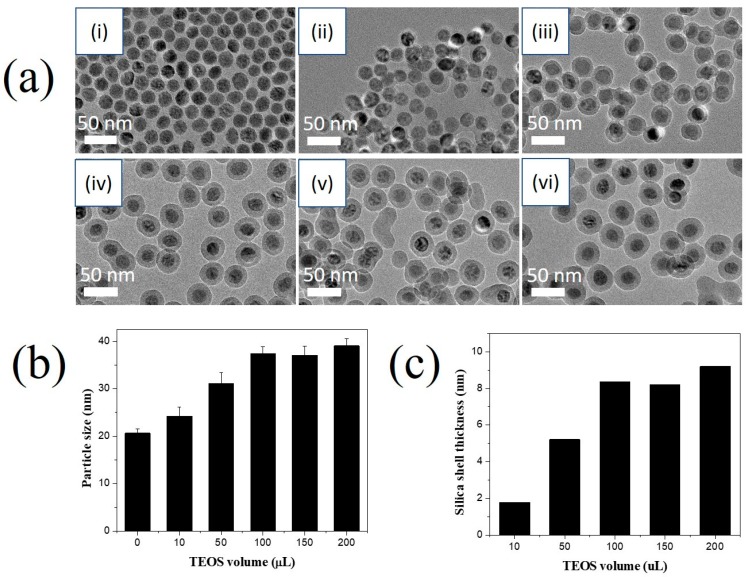
Effects of reverse microemulsion conditions on the silica coating of the magnetic nanoparticles. (**a**) TEM images (**b**) particle size and (**c**) silica shell thickness of MNP@SiO_2_ synthesized at different volume of TEOS: (i) 0 µL, (ii) 10 µL, (iii) 50 µL, (iv) 100 µL, (v) 150 µL, and (vi) 200 µLin the presence of 0.3 mL of Igepal^®^CO-520, 5 mg of MNP, and 100 µL of NH_4_OH.

**Figure 5 nanomaterials-10-00117-f005:**
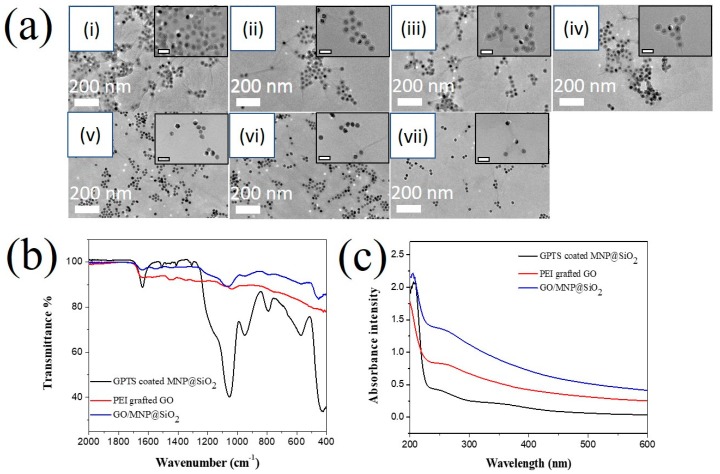
(**a**) TEM images of GO/MNP@SiO_2_ coated with various ratios of PEI grafted GO and GPTS-coated MNP@SiO_2_ (i) 2:1, (ii) 3:1, (iii) 4:1, (iv) 5:1, (v) 6:1, (vi) 8:1, and (vii) 10:1. The scale bar in inset is 50 nm. (**b**) Attenuated total reflection-Fourier transform infrared spectra and (**c**) UV–Vis spectra of GPTS-coated MNPs@SiO_2_, PEI grafted GO, and GO/MNPs@SiO_2_.

**Figure 6 nanomaterials-10-00117-f006:**
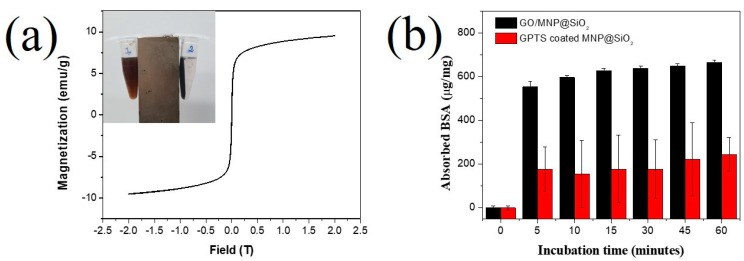
(**a**) Superparamagnetic hysteresis loop of GO/MNP@SiO_2_ used to determine saturation magnetization. The inset is photo of GPTS-coated MNP@SiO_2_ (left) and GO/MNP@SiO_2_ (right) when applying magnet for 5 min. (**b**) Adsorbed BSA per 1 mg of GO/MNP@SiO_2_ and GPTS-coated MNP@SiO_2_ incubated at various times.

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
