# Peer review of "Silica-Coated Magnetic Iron Oxide Nanoparticles Grafted onto Graphene Oxide for Protein Isolation"

_nanomaterials, 2020, doi:10.3390/nano10010117_

Round 1
Reviewer 1 Report
While revising their paper, the authors have taken into account the previously made comments.
Author Response
We appreciate the comment from the reviewer who spent invaluable time and effort. We have incorporated additional modifications based on the reviewers’ thoughtful comment, which have helped us to improve the manuscript.
Thank you for your help!
Reviewer 2 Report
Lines 17-19: As a result, the silica shell on the MNPs was optimized at 300 μL Igepal®CO-520, 5 mg MNP, 100 μL TEOS, 100 μL NH4OH and 3% of 3- glycidyloxypropyltrimethoxysilane (GPTS) to produce MNPs@SiO2-GPTS.“ must be transfered to the end of line 15 as following: „… iron oxide nanoparticles (MNPs). As a result, the silica shell on …“. Besides, the mentioned quantities can be also deleted in the abstract.
I would suggest to replace some sentences in the abstract with the lines 66-71 from the introduction section. Lines 66-71 is written very nice which give simply the reader an outlook on the developed procedure.
In line 20: is it MNP@SiO2 or MNPs@SiO2-GPTS?
Lines 48-52: „Zhang et al. Covalently … suspension [22].“. Please make it clear: who did what and in which reference. For example in the mentioned sentence, it is started with Zhang et al. and finished with [22] which is not related to each other. Please add suitable reference numbers at the right places. Please control it also throughout the manuscript.
Lines 59-61:“ To overcome the weak magnetic force of MNP@SiO2 and the difficult separation of GO, we develop a new nanomaterial by conjugating MNP@SiO2 to GO platform by covalent bonding in this study.“ And lines 23-25: „The quantity of BSA adsorbed onto 1 mg of GO/MNP@SiO2 increased sharply over time to reach 628 ± 24 9.3 μg/mg after 15 minutes, which 3.5-fold-higher than that of MNP@SiO2-GPTS.“. From these information, I learnd that two objectives were following in this study: 1) to magnetize GO by covalently binding MNP tot hem and 2) increasing the magnetic force of MNP@SiO2 by precisely controlling the SiO2 thickness layer. From my point of view, increased adsoption capacity is completely normal which happend due to the increased surface area. Please highlight in the abstract what is new in your study.
In my opinion, what you show in the inserted photo in figure 6a is facinating. However, it shows that covalently binding of GO to MNP increased the magnetic property and not the controlling the layer thickness. Am I right?
Line 124: „GO (2.5 mg in 0.5 mL)“. In 0.5 mL of what?
Line 124: „PEI solution (9.5 mL)“. What is the used solvent? Besides and also in line 75: Please give more detailed information about the used PEI. If it is a branched polymer or….
Line 124-125: How is the 40 µg/mL calculated?
Line 136-137: „An ethanolamine solution (100 μL, 1.0 M) was added to the 136 mixture and incubated for 1 h to block the unreacted epoxy groups“. How is this reaction?
Line: 142: „1% BSA (100 μL)“. Please make it clear, BSA in which solvent.
Line 150 and figure 1: In the text is mentioned epoxy group but in the figure –CHO is presenred, Why?
170-171: „The oleic acid that adsorbed on the surface of MNP during synthesis was exchanged to Igepal CO-520 in the cyclohexane.“ Did authors evaluate that or is just from other reports? Please add also oleic acid structre tot he figure 1.
171-173: „TEOS is then added to hydrolyze the water-oil interface and perform the ligand exchange between Igepal CO-520 adsorbed onto the MNP surface, which results in transfer of the MNPs into the water phase.“ The sentence is not clear for me. Please rewrite it to make it clearer and add suitable references.
Lines 173-174-: „Finally, the hydrolyzed TEOS on the MNP surface undergoes a condensation reaction to form the silica shell.“. The sentence is not clear for me. Please rewrite it to make it clearer and add suitable references.
In general, the step-by-step procedure used to coate MNPs is not clearly explained and is difficult to understand. Please rewrite it.
Line 225: „So, 100 μL of TEOS was chosen for further study.“. Why is this sentence here?
Lines 220-225: Have you evaluated the presence of the epoxy groups after this modfication? Please add the evaulted parameters at the right place. After reading the lines 274-276 I see that you did this evaluation.
Lines 235-236: „solutions of PEI“. Please write that which type of solvents have been used.
Figure 5a: In TEM images, I can just see MNPs. Where are GOs? I think it would be more clear if figure S2 (iii) and figure S3 (iv) are put by together in the manuscript.
Author Response
We appreciate the comment from the reviewer who spent invaluable time and effort. We have incorporated additional modifications based on the reviewers’ thoughtful comment, which have helped us to improve the manuscript.
Please see the attachment for our responses.

Reviewer 3 Report
In the manuscript entitled “Silica-coated magnetic iron oxide nanoparticles grafted onto graphene oxide for protein isolation” authored by Xuan-Hung Pham, Eunil Hahm, Hyung-Mo Kim, Byung Sung Son, Ahla Jo, Jaehyun An, Tuong An Tran Thi, Dinh Quan Nguyen and Bong-Hyun Jun synthesis of graphene oxide and silica coated magnetic particles conjugates is show. These conjugates were used for protein isolation. I don’t think that the presented manuscript is suitable for publication in Nanomaterials, mainly due to incremental character of the work.
The work might be regarded in two ways:
1) Main focus is in synthesis. But the synthesis itself is not very fascinating. There are no steps which are surprising (in sense of unusual utilization), difficult to perform, improving the procedure or new. While preparing the review I even made a note saying: “section 3.1 – optimization of synthesis is not very interesting for the story of the paper. It is important scientific part of the research but it might be moved to SI.” But later it appeared that this section in fact is a main part of the manuscript. Without it there will be almost nothing left.
2) Main focus is protein isolation. But there is only single (or maximum few) experiments shown in this respect. Authors claimed in several places that it might be used also for nucleic acids, but did not perform single experiment to support this. If this is the most important part of the paper than there is too little explanations given, e.g. on how hydrophilic/hydrophobic balance of the protein influence the separation, is it possible to in fact separate proteins from a matrix containing sugars, DNA and other biomolecules.
We are long past the time, where successfully preparing nanomaterials is enough. But in the case under consideration the addition of protein isolation is not well executed and looks like added at the very end of the work to increase chances of publication.
There are also some smaller issues with the paper:
Often amount added is denoted as concentration, which clearly is an error, e.g. P5 L167 – (in numerous other places as well). In Fig 3 MNP is given in uL and in Fig 4 in mg. This is confusing. Looking at Figure 1 I was wondering why PEI adsorbed on GO was first formed. I found answer not earlier than on page 7 “GO was dispersed in solutions 235 of PEI in our study to improve it stability aqueous solution”. Please Mark GPTS in Figure 1 similarly as PEI on GO (on third particles from left side). You discuss effect of Igepal and its hydrophilic groups – please show the structure. “Green” parts seems written by someone else and are sometimes of lesser quality. For instance P5 L190-194. The information on bare SiO2 particles is repeated later. Also should be Figure 3 and not Figures 3. “TEOS is then added to hydrolyze the water-oil interface and perform the ligand exchange between Igepal CO-520 adsorbed onto the MNP surface,” I have no clue what this sentence supposed to mean. Overall language needs some polishing (it is not bad, but there are some minor issues). For instance (but not only) Authors have tendency to use passive voice, where active voice would be much more suited.
Author Response
We appreciate the comment from the reviewer who spent invaluable time and effort. We have incorporated additional modifications based on the reviewers’ thoughtful comment, which have helped us to improve the manuscript.
Please see the attachment for our responses!

Round 2
Reviewer 3 Report
I appreciate Authors efforts to increase the quality of the manuscript. The work is designed in appropriate way and is executed according to the rules of good scientific practice. My main concern is still novelty. Again I divide my review in two parts.
Synthesis
Authors stated that “it is not easy to covalently conjugate MNP onto GO”. It might not be very easy, but it was done number of times in the past. I will give here just few examples
2011 Environmental Science & Technology “One-Pot Synthesis of Magnetic Graphene Nanocomposites Decorated with Core@Double-shell Nanoparticles for Fast Chromium Removal” – please see TOC and Figure 3. They did exactly what you aimed for. 2012 Chemical Engineering Journal “Synthesis, characterization, and adsorption properties of magnetic Fe3O4@graphene nanocomposite” see Figure 6. 2013 J. Mater. Chem. A, “Synthesis of water-soluble magnetic graphene nanocomposites for recyclable removal of heavy metal ions” Figure 8. 2013 Biomaterials “Graphene-based magnetic plasmonic nanocomposite for dual bioimaging and photothermal therapy” see inset in Figure 2.
There are numerous papers showing various methods on preparation of magnetic GO nanocomposites. I gave just 4 examples (notably, none of these examples were cited by you). The method presented by you is not novel. It is more of the engineering work to optimize the condition of the synthesis, but the synthetic protocols adopted by you was rather straightforward.
Protein isolation
You based your claims regarding possible application of the prepared nanocomposite for biomolecules isolation based on the literature data and support it with just single example of most commonly used BSA protein. BSA is a protein that is known to stick to various surfaces – e.g. in Western blot and ELISA it block the free surface of the nitrocellulose. It is also added to prevent adsorption on plastic surfaces of tubes in case of low concentration of biomolecules to be investigated. It is not surprising that it is scavenged by graphene based nanocomposite.
You cite publication “Fe3O4 nanoparticles on graphene oxide sheets for isolation and ultrasensitive amperometric detection of cancer biomarker proteins” (reference 20), but without any explanation. In this papers Authors show exactly the protein isolation. They used antibodies in addition to magnetic graphene oxide to add specificity to their approach.
There are also similar works:
2011 Proteomics “Facile preparation of magnetic graphene double‐sided mesoporous composites for the selective enrichment and analysis of endogenous peptides”
2014 Materials Science and Engineering C “Synthesis of magnetite/graphene oxide/chitosan composite and its application for protein adsorption”
To sum up, I still believe that the manuscript is not publishable in current form. You need to show other works on magnetic graphene composite and explain how you differ from them. Also I still believe that the isolation part should be the core of the paper, but this requires additional experiments.
Author Response
Thank you for your comment.
Please see the attachment.

This manuscript is a resubmission of an earlier submission. The following is a list of the peer review reports and author responses from that submission.
Round 1
Reviewer 1 Report
The authors describe the preparation of magnetic particles with a silica shell and carrying graphene oxide (GO). These particles display a higher saturation magnetization than could be achieved before. The particles are able to adsorb the protein bovine serum albumin (BSA) from a solution, and could most presumably adsorb other proteins as well.
Due to the fact that magnetic particles can easily be collected from solution with a magnetic separator, this approach allows the easy collection of for instance BSA from solutions.
The manuscript could be clarified as follows:
Line 74: Please indicate the chemical character of the magnetic particles that were purchased (iron oxide ? with …?)
Line 132:
“3 sizes of branched PEI (molecular weights of 600 (PEI 600), and 25,000 (PEI 25k) were investigated for GO grafting”
should this rather read as follows?:
“Two sizes of branched PEI (molecular weights of 600 (PEI 600), and 25,000 (PEI 25k) were investigated for GO grafting”
Line 159: it is not clear where the oleic acid on the MNP surface comes from. Please clarify.
Line 226: Please add a comment to this section about the easy collection of magnetic particles with a magnetic separator. This will emphasize the importance of the study, since the paragraph mentions that GO alone was not easily collected as a rationale/disadvantage (“However, the separation of GO from solution is a time-consuming and complicated process”).
Line 226 and elsewhere in the manuscript: The word “separation” may be misunderstood by readers that are involved in separation techniques, and interpreted as the separation of BSA from other proteins. Whereas the study mainly applies an adsorption of the protein material from a solution. Perhaps the word “isolation” better describes what has been done.
Line 239: Please replace absorbed by adsorbed, since a surface adsorption is meant.
Reviewer 2 Report
Abstract: currently abstract is written very poorly and needs to be rewritten stating clearly gap/need for the work followed by aim, methods, results and results significance.
Intro: similarly to abstract it is very difficult to see the gap and what the Team is trying to do and why? You write about all kinds of application but your modifications have application only to narrow area which should be the focus here. Clearly identified gap – why this work is important and needed – must be presented. And then just briefly what your aim is. You write all this steps and acronyms but I truly do not know why you do it, what you try to achieve and why your work has any value and significance?
Methods: is written in a way that experiments that are presented in methods can be repeated and contains required information. However many of the methods is not in method section, and incorporated elsewhere.
Some additional questions would be, what were your control samples? Was UV-Vis sufficient to assess separation? it seems to be rather low resolution method.
Major comments: characterisation methods are missing and they are wrongly embedded in result section.
Is statistical analysis missing?
Results: results often read as method section! Why do you write here all the steps for preparation of you material, these clearly belong to methods?
Since the manuscript is poorly written and organised with misplaced sections it is important first to address the structural issues and then evaluate the scientific merits which are at this stage very difficult to appreciate.
I do not see major value of this work, the significance is not present, control samples missing and statistical significance not considered.